# Harnessing Pharmacomultiomics for Precision Medicine in Diabetes: A Comprehensive Review

**DOI:** 10.3390/biomedicines13020447

**Published:** 2025-02-12

**Authors:** Dhoha Dhieb, Dana Mustafa, Maryam Hassiba, May Alasmar, Mohamed Haitham Elsayed, Ameer Musa, Mahmoud Zirie, Kholoud Bastaki

**Affiliations:** 1College of Pharmacy, QU Health, Qatar University, Doha P.O. Box 2713, Qatar; dhoha.dhieb@qu.edu.qa (D.D.); dana.nizar@qu.edu.qa (D.M.); m.hassiba@qu.edu.qa (M.H.); me1901975@student.qu.edu.qa (M.H.E.); 2Hamad Medical Corporation, Doha P.O. Box 3050, Qatar; mayelasmar.me@gmail.com (M.A.); mzirie@hamad.qa (M.Z.); 3College of Medicine, QU Health, Qatar University, Doha P.O. Box 2713, Qatar; am2204409@student.qu.edu.qa

**Keywords:** type 2 diabetes, precision medicine, pharmacomultiomics, pharmacogenomics, pharmacometabolomics, pharmacoproteomics, pharmacotranscriptomics, pharmacomicrobiome, pharmacoepigenomics, personalized therapy

## Abstract

Type 2 diabetes (T2D) is the fastest-growing non-communicable disease worldwide, accounting for around 90% of all diabetes cases and imposing a significant health burden globally. Due to its phenotypic heterogeneity and composite genetic underpinnings, T2D requires a precision medicine approach personalized to individual molecular profiles, thereby shifting away from the traditional “one-size-fits-all” medical methods. This review advocates for a thorough pharmacomultiomics approach to enhance precision medicine for T2D. It emphasizes personalized treatment strategies that enhance treatment efficacy while minimizing adverse effects by integrating data from genomics, proteomics, metabolomics, transcriptomics, microbiomics, and epigenomics. We summarize key findings on candidate genes impacting diabetic medication responses and explore the potential of pharmacometabolomics in predicting drug efficacy. The role of pharmacoproteomics in prognosis and discovering new therapeutic targets is discussed, along with transcriptomics’ contribution to understanding T2D pathophysiology. Additionally, pharmacomicrobiomics is explored to understand gut microbiota interactions with antidiabetic drugs. Emerging evidence on utilizing epigenomic profiles in improving drug efficacy and personalized treatment is also reviewed, illustrating their implications in personalized medicine. In this paper, we discuss the integration of these layers of omics data, examining recently developed paradigms that leverage complex data to deepen our understanding of diabetes. Such integrative approaches advance precision medicine strategies to tackle the disease by better understanding its complex biology.

## 1. Introduction

Diabetes mellitus (DM) is one of the most common chronic metabolic diseases worldwide with an alarmingly rapid increase in incidence over the years [1,2]. According to 2021 data, approximately 537 million people suffer from diabetes, with the highest comparative prevalence to the world population of diabetes (18.1%) being in the Middle Eastern and North African (MENA) region [1]. Moreover, the International Diabetes Federation (IDF) predicts an increase in DM of around 46% by 2045, reaching an astounding number of 783 million diabetic patients globally [1]. Qatar exemplifies this global trend, with IDF data indicating a diabetes prevalence of 19.5% in 2021, expected to increase to 22.8% by 2045 [3]. Furthermore, Soliman et al. underscored that the Arab region exhibits a prevalence of diabetes that exceeds the global average. Remarkably, five of the ten countries with the highest diabetes prevalence among adults are located in the Arab Gulf region, including Qatar, which reports a prevalence rate of 20.2% [4].

DM, particularly T2D, contributes to a significant global health and economic burden [5]. The economic burden of total diabetes related health expenditure is substantial, projected to increase from USD 966,000 in 2021 to USD 1,053,000 in 2045 [1], straining healthcare systems and affecting the delivery of quality care. The economic implications are paralleled by the clinical challenges associated with T2D, which accounts for approximately 90% of all diabetes cases. In 2019 alone, direct healthcare costs attributable to T2D reached an astounding USD 760 billion globally [6].

Despite the heterogenous nature of DM, its diagnostic criteria, as defined by the American Diabetes Association (ADA), are limited to measures such as glycated hemoglobin (HbA1c), fasting plasma glucose, the oral glucose tolerance test (OGTT) and/or random glucose readings [7]. These tests assess impaired glucose control but do not address disease etiology, classification, underlying pathologies, socioeconomic factors, psychosocial factors, or underlying individual genomic predisposition [8,9], limiting effective diagnosis, prevention, and tailored therapy. Despite the aforementioned heterogeneity, treatment is almost standard for many DM patients without accounting for ‘personalized’ factors [10], and there is substantial interindividual variability in drug response [11]. Standard therapeutic approaches, including a variety of non-insulin antidiabetic drugs such as biguanides, sulfonylureas, and SGLT2 inhibitors, often fail to achieve recommended level of glycemic response [7,12]. This is evidenced by data from the U.K. General Practice Research Database showing that only about 50% of the patients initiating treatment with agents like metformin achieve target HbA1c levels of <7% [13]. The factors that account for this variation are not well understood. Given the heterogeneity in patient responses and sometimes serious adverse drug reactions (ADRs), which account for a significant proportion of hospital admissions [14,15], there is a compelling need for a shift towards personalized medicine in the management of T2D. The challenging nature of DM etiology, pathogenesis, and disease progression among patients can be addressed by a more comprehensive and personalized approach. Precision medicine as defined by the Food and Drug Administration (FDA) involves tailored prevention and treatment strategies to optimize care in a timely and effective manner [16]. It presents a unique potential in tackling the complex and polygenic nature of DM, specifically T2D management by utilizing detailed knowledge of the patient’s genetic and other omics data to tailor diagnostics, prevention strategies, prognostics, and therapeutics interventions, thereby avoiding the inefficiencies and inconveniences associated with traditional trial-and-error methods [17]. The Precision Medicine in Diabetes Initiative (PMDI) was established in 2018 by the American Diabetes Association (ADA) in partnership with the European Association for the Study of Diabetes (EASD). The ADA/EASD PMDI includes global thought leaders in precision diabetes medicine who are working to address the burgeoning need for better diabetes prevention and care through precision medicine [18,19]. Central to this approach are multiomics data, including pharmacogenomics, pharmacotranscriptomics, pharmacoproteomics, pharmacometabolomics, pharmacomicrobiomics, and epigenetics. Collectively, the integration of this variety of biological layers of data into clinical practice offers transformative potential. This multi-layered strategy embodies a forward-thinking paradigm, advancing both understanding and treatment in a personalized and efficacious manner that enhances efficacy while minimizing ADRs (Figure 1). In this review, we aim to explore and discuss the applications of these various pharmacomultiomics approaches in the context of T2D. We will discuss their roles in enhancing personalized treatment strategies, predicting drug responses, identifying prognostic targets, and elucidating complex drug–gene interactions. Through this comprehensive evaluation, we seek to underscore the significance of integrating multiomics strategies to enhance personalized medicine in the treatment of T2D.

## 2. Enhancing Personalized Medicine in T2D Through Pharmacogenomic Insights

In the management of T2D, the efficacy and safety of drug responses exhibit considerable interindividual variability, largely attributable to differences in genetic makeup across diverse population groups [11]. Traditional drug dosages and therapeutic windows are typically calculated based on an empirical ‘population average dose’ [20], which does not account for individual variability. This can lead to over- or underdosing in real-life practice regimens due to the genetic variability. Pharmacogenomics is the discipline that investigates how our entire genome influences individual responses to drugs, and, more specifically, pharmacogenetics focuses on genetic variation at a population level, and how these variants can affect therapeutic outcomes and the incidence of adverse effects [21]. Significant advancements in personalized medicine in T2D have been driven by comprehensive genomic analyses, as demonstrated by research from the Metformin Genetics (MetGen) Consortium (https://pgrn2016.weebly.com/metgen.html, access date 12 December 2024). This research identified a link between the SNP rs8192675 within the ***SLC2A2*** gene and variations in glycemic response to metformin, showcasing how genomic insights can refine patient management [22]. Further substantiating the clinical relevance of pharmacogenomics, the first Genome-Wide Association Study (GWAS) on metformin response, involving subgroups from the GoDARTS cohort and the UK Prospective Diabetes Study (UKPDS), identified the C allele of rs11212617 [23]. This allele is linked to improved glycemic control under metformin treatment, demonstrating the value of genomic approaches in optimizing therapeutic protocols. Moreover, exploration of genetic variants in ***TCF7L2***, ***PPARG***, ***KCNJ11***, ***WFS1***, ***SLC30A8***, ***JAZF1***, and ***HNF1B*** has shed light on their significant role in influencing the risk of developing T2D. This expanding genomic database not only enhances our understanding of disease mechanisms but also supports the development of targeted prevention strategies, transitioning from traditional treatment methods to more personalized interventions. Systematic reviews focusing on drug response and genetic associations have uncovered significant drug–gene interactions influencing glycemic control. These findings relate to various medications including metformin, sulfonylureas, repaglinide, pioglitazone, rosiglitazone, and acarbose, highlighting a range of influential loci [24]. Metformin has been extensively studied for pharmacogenomic interactions concerning T2D. Notable findings discovered that the rs622342 SNP in the ***SLC22A1*** gene relates to changes in HbA1C levels post-treatment, needing further replication [25]. Concurrently, the ***SLC47A1*** gene’s rs2289669 non-coding polymorphism has been linked to a HbA1C reduction per minor A allele versus the G allele [25]. Additionally, ***ATM*** gene’s rs11212617 polymorphism, pivotal for cell-cycle control and DNA repair, has been observed to correlate with metformin responsiveness [26], although results vary significantly across ethnicities, with no impact noted in an Iranian T2D cohort [27,28,29,30]. Other critical studies found that the rs122083571 and rs72552763 polymorphisms in ***OCT1*** could quadruple the risk of metformin intolerance [31]. Contrasting responses were also noted within Indian populations for the rs2297374 and ***SLC22A1*** rs622342 polymorphisms, potentially affecting metformin efficacy [30,32]. Further research on Mexican subjects identified associations between ***SLC22A1*** CC-rs622342, AA-rs628031, and GG-rs594709 variants and reduced metformin effectiveness, marked by increased HbA1c levels [33]. The ***SLC22A3*** gene and its SNPs (rs12194182, rs2292334, rs2504927, rs3123634), however, showed no association with metformin action in Caucasian subjects [34]. Conversely, specific variants in ***TCF7L2*** and ***PRPF31***, ***CPA6***, ***STAT3*** genes have demonstrated novel interactions with metformin’s glucose-lowering mechanisms, potentially influencing clinical outcomes [35,36]. Additionally, specific ***SLC29A4*** polymorphisms have been implicated in metformin absorption, influencing gastrointestinal tolerance [37]. Furthermore, the complex genetic landscape governing the response to other antidiabetic medications like thiazolidinediones and sulfonylurea has been similarly elucidated through various studies. For instance, the adiponectin gene (ADIPOQ) variants have shown changes in fasting glucose and HbA1C levels after rosiglitazone treatment [38]. The crucial role of ***CYP2C8*** gene variants in modulating the clearance rate of rosiglitazone further underscores the nuanced interactions between genetic factors and drug metabolism [39]. Thiazolidinediones, recognized as ***PPAR*** activators, notably influence metabolic pathways by reducing circulating free fatty acids and enhancing insulin sensitivity, thereby mitigating hyperglycemic episodes [40]. Genetic variations like the rs296766 T allele of ***AQP2*** and rs12904216 G of ***SLC12A1*** have been associated with edema in users of rosiglitazone, emphasizing the genetic predisposition to side effects [41]. Further studies have pinpointed polymorphisms such as PPARG C1A Thr394Thr and Gly482Ser and the P12A variant in ***PPARG*** as critical determinants of the therapeutic response to thiazolidinediones in various ethnic populations [38,42,43]. Regarding sulfonylureas, the ***KCNJ11*** E23K variant has been identified as a significant factor influencing drug efficacy, particularly in Caucasian populations [44]. T2D patients carrying specific polymorphisms like rs7903146 and rs12255372 in the ***TCF7L2*** gene have been observed to exhibit poor responses to sulfonylurea treatments, highlighting the potential for genetic screening in optimizing therapeutic approaches [45,46,47,48]. The exploration of DPP4 inhibitors and ***GLP1*** analogs in T2D treatment also reveals significant genetic influences. Variants such as rs6923761 in the ***GLP1R*** gene have demonstrated altered glucose control responses in Central European T2D cohorts exposed to gliptins [49]. Similarly, polymorphisms within ***DPP4***, like rs6741949, show correlations with insulin secretion and glucose tolerance, suggesting personalized treatment approaches could optimize the benefits of these drugs [50]. The complex interaction of SGLT2 inhibitors with genetic variants has shown mixed results across studies. While some ***SGLT2*** gene loci variants exhibit no clear association with drug response, others, like the UGT1A9*3 and UGT2B4*2 polymorphisms, have increased plasma concentrations of canagliflozin, indicating variable patient responses based on genetic makeup [51,52]. For detailed exposition of some of the genes involved, their significant genetic polymorphisms, and respective clinical implications associated with some of the major antidiabetic drugs, refer to Table 1.

## 3. Pharmacometabolomics in T2D for Advancing Drug Response

Pharmacometabolomic research seeks to explain the impact of an individual’s metabolic profile (metabotype), identified through metabolomics, on their drug responses, refs. [53,54] positioning it as a vital tool in precision diabetes medicine. Metabolomics, by analyzing small molecules integral to metabolomic process and cell signaling, brings us closer to representing the clinical phenotype of type 2 diabetes and has revamped biomarker discovery, disease subtype/clustering and disease pathogenesis understanding [55,56].

Despite challenges such as low coverage, unknown metabolite identification, low throughput, and difficulties in interpretation due the vast chemical diversity [57], pharmacometabolomics can be particularly effective in complex diseases such as DM as similar phenotypes can be affected by multiple different pathophysiologic processes. It also aids in discovering novel biomarkers for diseases as well as identifying off-target side effects in marketed drugs and newly developed chemical entities [54]. Metabolic biomarkers act as a direct signature of biological functions which aids in tracking back to the phenotype [58]. The most common profiling technology for identifying metabolic biomarkers is liquid chromatography–mass spectrometry (LC-MS), which illustrates alterations in the metabolic profile and specific metabolic abnormalities by quantifying and specifying the number of modified biomarkers [59,60]. Metabolic patterns of individuals can be used for disease diagnosis and predicting future illnesses and response to therapy. Multiple pharmacometabolomic studies have also been conducted on certain anti-diabetic medications assessing individuals’ varied responses to these drugs. For instance, a pharmacometabolomic study conducted on metformin to differentiate good and poor responders has concluded that an increased level of certain metabolites including sphingomyelins, acylcholines, and glutathione was found in good responders, while poor responders showed high levels of metabolites resulting from glucose metabolism and gut microbiota metabolites [61]. A cross-sectional study using targeted metabolomics data from the Nightingale platform across four Dutch cohorts examined metabolic associations with glycemic control stratified by HbA1c levels and glucose-lowering medications (metformin, sulfonylurea, insulin). Analysis showed that 26 out of 162 metabolites notably correlated with poor glycemic control, particularly between glutamine and BCAA/aromatic amino acids [62]. This study and others highlight metformin’s extensive impact on metabolic pathways including the TCA cycle, glucose, and lipid metabolism, significantly altering metabolites such as hydroxyl-methyl uracil and glycerol-phospholipids [63,64]. Furthermore, the Copenhagen Insulin and Metformin Therapy trial linked metformin therapy to reduced levels of certain amino acids which are associated with insulin resistance and mitochondrial dysfunction, although the specific metabolites predicting HbA1c reduction were not identified [65]. Treatment with gliclazide-modified release improved insulin sensitivity and altered metabolic pathways such as TCA and ketone body metabolism, indicative of its efficacy in T2D management [66]. To further improve the utility of metabolomics, we propose that future studies should try to monitor metabolic dynamics/rates with metabolic tracers, known as fluxomics [38,39].

## 4. Pharmacoproteomics in T2D: Unraveling Prognostic Biomarkers and Novel Therapeutic Targets

Proteomics encompasses exploring comprehensive protein structures, functions, interactions, composition, and cellular activities [67]. Proteomes are significantly used to investigate different protein expressions and modifications that will affect the body’s biological processes. Regarding pharmacological research, pharmacoproteomic is known as an important approach to understanding the drug mechanism of action, side effects, toxicity, drug resistance, and new drug target discovery [68]. It can also be used to categorize patients according to their protein analysis using different bioinformatics techniques. Pharmacoproteomic shows an increased functional representation of patient-to-patient variation and can act as the linkage between genotype and phenotype due to the significant post-translational modification taking place during protein synthesis [69]. Given that the majority of current drug targets are proteins, understanding protein expression and mRNA conformational changes can aid in categorizing patients as responders or non-responders. This classification process can ultimately advance personalized drug discovery and facilitate the identification of new targets including allele-specific targets [70]. In DM, Proteomic data serve as valuable clinical markers to monitor patients’ progress, enable easier genetic screening, and address the gaps in molecular medicine [71]. Proteomics present a valuable tool in understanding the pathogenicity of diseases and the discovery of potential biomarkers such as MASP [71], Ftuin-A [72], and others. However, the implication of these biomarkers in diagnostic and monitoring techniques is not routinely used in clinical settings. One potential pathway resides in constructing a model that incorporates multiple serum biomarkers to improve the potential for detection, diagnosis, and prognosis of T2D. Nonetheless, translating these findings into routine clinical tests faces challenges, particularly in characterizing diverse protein profiles, especially those of low abundance [73]. Further studies on a larger scale can validate the reliability and validity of this model, which can help in better management of T2D through advancing effective diagnostic and prognostic tools [74]. A recent study showed that the proteomic approach serves a large number of individuals with isolated impaired glucose tolerance (iIGT) who are only detectable through oral glucose tolerance tests (OGTTs). It concluded that adding only three proteins (RTN4R, CBPM, and GHR) to the best clinical model significantly improved discrimination, as higher plasma protein levels of these proteins increase the future risk of T2D [75]. Moreover, pharmacoproteomic approach offers valuable understanding of treatment mechanism of action, potential drug targets, and medication toxicities in T2D. For instance, a study conducted on diabetic mice using peroxisome proliferator activator receptor (PPAR) showed that by analyzing protein expression profiles post-treatment, biochemical pathways affected by the drug were discovered, which gives important insight into new therapeutic targets. The study results revealed alterations in the liver proteins that affect the fatty acid metabolism, shedding light on drug mechanisms and potential therapeutic effects. In addition, proteomic screening in the study has identified drug-induced toxicity, represented by proteins associated with lipid accumulation in the livers of rats treated with certain hypoglycemic agents [76].

## 5. Pharmacotranscriptomics in Diabetes: Exploring Gene Expression Dynamics for Personalized Treatment Strategies

Pharmacotranscriptomics in diabetes offers profound insights into the gene expression dynamics integral to developing personalized treatment strategies. Transcriptomics, focusing on the complete set of RNA transcriptions within an organism, tissue, or cell, has significantly advanced diabetes management [77]. As Mohan and Radha (2019) highlighted, integrating various omics, including transcriptomics, could shift diabetes treatment from a one-size-fits-all approach towards more precise management, potentially curtailing diabetes complications and enhancing patient quality of life [78]. In exploring the intricate pathogenesis of T2D, transcriptomic studies have shed light on the roles of immune inflammation and lipid metabolism. These studies provide supplementary data on pathways affected by T2D, such as the ubiquitin–proteasome system and cell-cycle pathways [79]. Environmental factors further modulate gene expression, thereby influencing phenotypes and metabolic disease risks [80]. Studies have primarily focused on islets and peripheral tissues, including the liver, muscle, and adipose tissue, using tools like oligonucleotide microarrays and RNA sequencing (RNA-seq) [81]. These tools have unveiled transcriptional differences across these tissues, enhancing our understanding of T2D [79]. The translational impact of transcriptomics is exemplified in the study of drug responses, particularly metformin, where genetic variations alone fail to fully explain the variability in patient responses. Exploratory transcriptomic analyses have identified metformin’s broad pharmacological effects, shedding light on potential therapeutic biomarkers [82]. Recent studies demonstrate metformin-induced transcriptional changes in animal models, though similar human studies are sparse. For instance, Guo et al. observed metformin-related changes in coding and non-coding RNA profiles in mice [83,84,85]. Other analyses have linked metformin treatment to calorie restriction-like transcriptomes and distinct gene expression profiles related to cardiovascular benefits. In vitro studies of cell lines, such as adrenal H295R and MCF7, have associated metformin with processes in energy metabolism and cancer pathways [86,87]. Whole-transcriptome analysis using total RNA sequencing in healthy individuals on metformin has revealed its effects on gene expression, notably genes related to the immune response, insulin production, and cholesterol homeostasis. This study marks the first to assess metformin’s immediate impact on global gene expression in healthy subjects, unveiling the drug’s association with the intestinal immune system and gene clusters showing subject-specific expression [88,89]. Furthermore, RNA-Seq has identified various novel targets of metformin, such as enrichment in transcriptional regulators like FOXO3a in human fibroblasts and modulated gene expression that suggests uncharacterized non-genetic effects [88,90]. While genetic inheritance partly explains metformin response heterogeneity, recent studies indicate patient- and cell type-specific responses, signaling gaps in our understanding that pharmacotranscriptomics seeks to fill [88]. The transcriptomic analysis extends to drugs like thiazolidinediones, targeting PPARγ to counter insulin resistance in T2D. Varying patient responses, underscored by genetic variations, have been highlighted in studies using human adipose stem cells. Such studies demonstrate inconsistent gene activation in response to rosiglitazone due to specific SNPs, emphasizing the necessity for personalized approaches. CRISPR–Cas9 interventions have highlighted the potential to modify gene responsiveness, translating findings from cell models to clinical applications. These insights underline the role of genetic variations in antidiabetic therapy outcomes and propose pharmacotranscriptomics as a frontier for developing personalized diabetes treatments [91].

## 6. Pharmacomicrobiomics for Navigating the Microbial Influences on Drug Response in Diabetes Management

Pharmacomicrobiomics is a field of study that combines our understanding of microbiome variations (i.e., genetic composition and metabolic activity) and how those variations among individuals affect the response and toxicity of medications [92]. Microbiome is a term that encapsulates a community of microorganisms residing in a specific location, their genetic makeup, and the interaction of that community among itself and its environment [93]. The emerging tools of next generation sequencing are high-throughput technologies that enable profiling large cohorts of the microbiome and the organisms hosting it [94]. This allowed the building of various microbial catalogues [95,96,97] and databases such as The Integrative Human Microbiome Project (iHMP) [98]. Integrating these tools with the clinical observations of patients’ drug responses enhances our pharmacomicrobiomics knowledge and paves the way for personalized drug choices. The microbiome composition can be affected by ingested drugs, and the pharmacokinetics of these drugs can also be influenced by the gut microbiome metabolizing them. This bi-directional relationship between medications and the individuals’ microbiome should be accounted for during drug designs [99], including the design of antidiabetic drugs. The benefits of viewing antidiabetic drugs through the lens of microbiota have already been shown by various studies investigating anti-diabetic drugs intolerance and responsiveness. Pharmacomicrobiomics can help in enhancing drug tolerance, such as in the case of Metformin. This oral biguanide medication is a common effective anti-diabetic drug [100]. However, its adverse side effects can affect the life quality of certain patients or in some cases lead to the discontinuation of the treatment hindering its benefits. Many studies regarding metformin gastrointestinal side effects drew a link between drug intolerance and the gut microbiota composition [101]. Therefore, intervening with the gut microbiota is one possible approach of resolving drug intolerance. Probiotic intake is an example of an intervention method that has been associated with fewer gastrointestinal side effects when combined with metformin [102]. Furthermore, differences in pharmacokinetics among two ethnicities, partly attributed to gut microbiota variations, have been found to lead to optimal clinical outcomes at different doses of an anti-diabetic supplement [103]. Understanding pharmacokinetics in different cohorts in regard to anti-diabetic drugs can help in finding methods to intervene or lower the dosage when possible, to alleviate unnecessary adverse effects. In addition, pharmacomicrobiomics aids in understanding variabilities in drug responsiveness. Bioaccumulation, microbial resistance, and isozymes are all factors in which the gut microbiota can affect the drug, and, in return, the host’s responsiveness. Bioaccumulation is the process in which the drug is taken into the bacterial cell without metabolizing it. This could possibly affect its availability for the host [103]. Bioaccumulation has been reported in pre-clinical studies for the diabetes drug rosiglitazone [104]. Furthermore, microbial resistance has been identified in the case of acarbose. Patients taking this alpha-glucosidase inhibitor and harboring bacterial genes that can inactivate or degrade acarbose have been associated with poor drug response toward it [104,105]. Moreover, human microbial isozymes can affect drug efficacy. This is exemplified by sitagliptin, a DPP-4 inhibitor used in diabetes treatment. However, the drug does not inhibit the microbial DPP-4, which leads to drug response variability [106]. Gut composition can also be altered by other factors such as antibiotic intake, leading, in return, to lower drug efficacy [107]. Taking into consideration gut composition variability and the varying ways it can interact with the drugs can broaden the medication’s efficacy.

## 7. Pharmacoepigenomics in Diabetes: Exploring Epigenetic Influences on Drug Efficacy and Personalized Treatment

Pharmacoepigenetics, an emerging branch of pharmacology, examines how epigenetic modifications, including DNA methylation, histone modifications, and non-coding RNA, affect individual responses to drugs; however, this domain remains less understood [108]. This field is not only concerned with blood-based epigenetic biomarkers predicting therapeutic response but also includes therapy-induced epigenetic changes and the potential use of epigenetic therapies, such as inhibitors targeting epigenetic enzymes. Recent research highlights the utility of blood-based epigenetic biomarkers in predicting the glycemic response and tolerance to metformin in newly diagnosed T2D patients [108]. Methylation of 11 specific CpG sites was associated with a non-response to metformin, while methylation of four other sites linked to intolerance. These methylation risk scores (MRS) successfully differentiated between responders and non-responders, as well as tolerant and intolerant individuals across multiple cohorts, suggesting a promising avenue for precision medicine. Furthermore, metformin has been shown to affect DNA methylation of genes encoding transporters like OCT1, OCT3, and MATE1 in the liver of individuals with T2D [109]. Short-term exposure also altered DNA methylation in non-diabetic individuals [110]. Incretin drugs, such as GLP1R agonists, demonstrated the ability to prevent glucose-induced methylation changes in genes like NFKB1 and SOD2 within endothelial cells, hinting at potential benefits for vascular complications [111]. These drugs even reversed epigenetic changes in animal models exposed to adverse intrauterine conditions [112]. Moreover, statin therapy has been implicated in differential DNA methylation patterns in both diabetic and non-diabetic individuals [113,114]. Notably, changes were observed in genes related to cholesterol metabolism, such as ABCG1, DHCR24, and SC4MOL. Causal mediation analyses suggest that these methylation changes may underlie some of the metabolic effects attributed to statins [113,114]. Epigenetic therapies, which include inhibitors of DNA methyltransferases (DNMTs) and histone deacetylases (HDACs), represent another promising avenue, although they are primarily employed in cancer treatment [115]. Dysregulation of epigenetic enzymes in diabetes, such as increased DNMT3B in myotubes and decreased TET1 in adipose tissue, suggests their potential role in T2D management [116,117,118,119,120]. Moreover, exposure to diabetogenic conditions affected the expression of several DNMTs in pancreatic islets [121]. HDAC inhibitors have shown promising results in improving beta-cell function and insulin secretion by reversing adverse epigenetic modifications [118,122,123,124,125]. Current findings underscore the transformative potential of pharmacoepigenetics in personalizing diabetes treatment, guiding therapeutic interventions, and perhaps introducing novel therapies that target specific epigenetic mechanisms.

## 8. Standardizing Muli-Omics Data Integration in T2D

Efficient integration of multiomics datasets is fundamental for elucidating the molecular and genetic landscapes of T2D, thereby facilitating the advancement of precision medicine. Current efforts are directed towards this goal by employing cutting-edge technologies, including machine learning (ML) and artificial intelligence (AI), to effectively analyze and integrate large-scale omics data. For example, the application of supervised ML models for integrating DNA methylation, RNA sequencing, SNPs, and phenotypic data using frameworks such as DIABLO, which employs Partial Least Squares (PLS) for data integration, has demonstrated considerable success. This methodology achieves notable predictive accuracy and aids in biomarker discovery, thereby enhancing the precision of diagnoses and understanding of disease etiology [126]. Moreover, AI applications in the realm of multiomics allow for unbiased evaluations of integration tools like MOFA+ and GFA, which are assessed for their efficacy in feature selection across extensive biological datasets. These tools, through unsupervised learning models, have proven effective in differentiating between patient and control samples, highlighting AI’s critical utility in refining feature selection to accurately reflect underlying biological variations in T2D [127]. As the field of T2D research evolves, integrating advanced machine learning (ML) strategies with multiomics data is poised to play a transformative role in our comprehension and management of chronic diseases. This convergence not only facilitates a deeper understanding of genotype-phenotype relationships but also enhances the precision of disease classification, prognostication, and therapeutic interventions. Despite the promising potential of integrating AI and ML with multiomics, several challenges persist. The protection of data privacy and security is paramount, along with the ethical use and sharing of sensitive biological information [128]. The complexity and variability of genetic and phenotypic data require sophisticated algorithms and larger computational resources to achieve accurate and timely predictions. Overfitting, where models adapt too narrowly to training data, and underfitting, where models fail to capture data complexities, remain critical issues that require careful model development and validation. Furthermore, the absence of regulatory frameworks to oversee the ethical and responsible use of AI in healthcare adds another layer of complexity. While AI-driven systems are currently permitted as adjunct tools, a regulatory transition to support their full-fledged adoption in clinical decision-making could enhance their utility.

## 9. Precision Medicine Initiatives in T2D

The era of precision medicine in T2D has been significantly propelled by advancements following the sequencing of the human genome. These technological developments have enabled the large-scale acquisition of genetic and multi-dimensional biological data, facilitating the discovery of novel biomarkers that offer insights into the disease’s intricacies. These breakthroughs have catalyzed the formation of global initiatives aimed at leveraging existing biobanks and collaborative networks to stratify patient populations, thereby enhancing treatment efficacy through tailored therapeutic strategies. In Europe, the Innovative Medicines Initiative (IMI), a EUR 5.6 billion program launched in 2008, exemplifies such efforts through its focus on major diseases, including T2D. Projects funded by IMI, like SUMMIT, DIRECT [129], RHAPSODY, and BEAt-DKD, are dedicated to advancing biomarker discovery and validation to enable precise patient stratification and, in some instances, diagnostic reclassification by integrating data such as genotypes and metabotypes [130,131]. The UK Biobank (https://www.ukbiobank.ac.uk, accessed on 24 October 2024), established in 2005, has played a pivotal role by offering rich genetic, metabolic, and lifestyle datasets, supplemented with phenotypic data like MRI scans. This resource enhances the understanding of gene-environment interactions relevant to diabetes, despite challenges posed by potential confounding factors [132]. In the United States, initiatives such as the Precision Medicine Initiative (PMI), announced in 2015 [133,134], the Million Veteran Program [135], and the Accelerating Medicines Partnership in T2D [136], focus on assembling comprehensive health data and fostering public-private partnerships for advancing T2D research. Platforms like the T2D Knowledge Portal exemplify these efforts, providing a central repository for data sharing and analysis. In the Nordic region, the Nordic Precision Medicine Initiative and projects like the Estonian Genome Center and FinnGen endeavor to integrate genomic insights into clinical practice. In Iceland, deCODE genetics significantly contributes to understanding T2D genetics, ensuring robust drug target validation [137]. Globally, projects such as the Saudi Human Genome Program (SHGP) [138] and China’s National Precision Medicine Initiative [139] are dedicated to leveraging genomic data to address prevalent diseases like T2D. These initiatives aim to sequence extensive populations, develop disease-specific cohorts, and promote national-scale genomic medicine, reflecting substantial governmental investments. These worldwide endeavors highlight a shared commitment to translating genomic discoveries into clinical practice, paving the way for more precise management and treatment of T2D.

## 10. Limitations

This review highlights key aspects of precision medicine for T2D but acknowledges several limitations. Firstly, our focus on a selective scope of literature may omit broader perspectives, including other impactful areas like behavioral interventions and lifestyle modifications. The field is challenged by high costs, ethical concerns, data security issues, and a shortage of skilled professionals for advanced algorithm development. The necessity and value of such complexity for diabetes management remain uncertain. Economic implications of implementing precision medicine are also not explored, which is crucial for assessing real-world applicability. Moreover, the review notes a lack of large-scale, high-quality studies and randomized trials needed to validate precision medicine’s benefits. Many studies focus narrowly on blood glucose prediction, overlooking a comprehensive patient view. Sampling biases and issues with patient adherence further limit study validity and generalizability. While we provide an overview of multiomics approaches, the depth of computational data integration methods is limited, an area we plan to expand on. Additionally, we do not discuss pre-emptive genotyping for reducing adverse drug reactions, which is a proven strategy. It is also noteworthy that our discourse may not fully encompass the integration of precision medicine within existing clinical workflows, potentially underrepresenting the logistical obstacles and potential resistance encountered from healthcare providers. Addressing these gaps and incorporating broader perspectives will strengthen future research, ensuring precision medicine strategies in T2D are scalable, economically viable, and applicable across diverse populations.

## 11. Conclusions

The integration of different pharmacomultiomics data offers significant potential for enhancing precision medicine in the management of T2D. This multiomics approach facilitates a deeper understanding of the disease and advances translational research and clinical practice by enabling more personalized and effective treatment strategies. Essential objectives such as patient stratification for optimized therapy selection and a comprehensive understanding of clinical phenotypes are addressed, adding a complex layer beyond traditional clinical profiles. These strategies focus on achieving patient-centered outcomes by tailoring treatment regimens to individual needs. Nonetheless, translating these innovative approaches into clinical practice presents considerations that require attention, including thorough economic evaluations, addressing health disparities, and ensuring compliance with regulatory standards. Furthermore, successful implementation demands comprehensive stakeholder education, rigorous assessments of real-world evidence, proactive regulatory engagement, and enhanced research infrastructure. Moving forward, it is crucial to establish comprehensive policy frameworks that responsibly address ethical, legal, and social concerns to integrate these technologies effectively into healthcare systems (Figure 2). It is essential to continue exploring these layers of omics to refine therapeutic targets, increase drug efficacy, and decrease adverse effects associated with T2D treatments. Embracing these strategies will lead to more precise, effective, and patient-specific therapeutic outcomes, ultimately improving the prognosis and quality of life for patients with T2D. More efforts in the future should focus on converting these findings into clinical practice and developing policies to support the integration of precision medicine into healthcare systems globally.

## Figures and Tables

**Figure 1 biomedicines-13-00447-f001:**
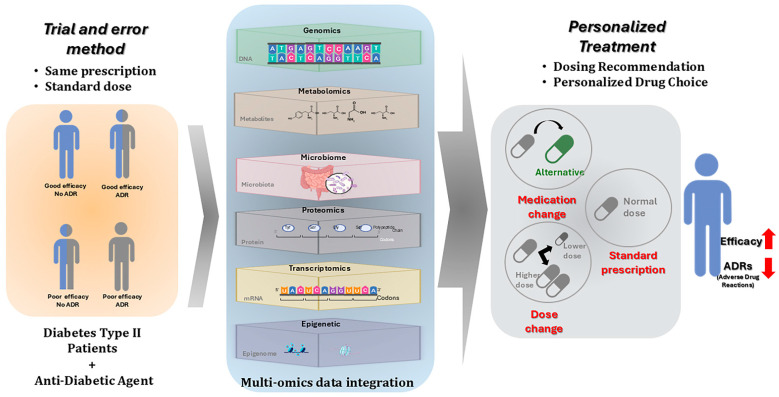
Advancing T2D management through multiomics integration in personalized treatment versus standard therapeutic approaches.

**Figure 2 biomedicines-13-00447-f002:**
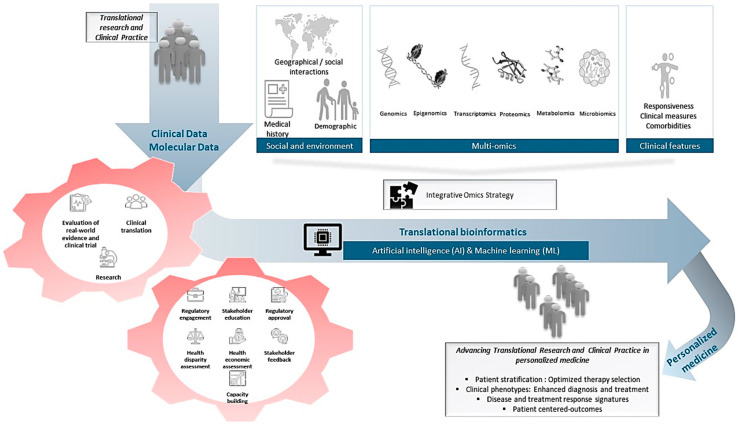
Integrative omics strategy in precision medicine.

**Table 1 biomedicines-13-00447-t001:** Genetic polymorphisms and pharmacological properties of some major antidiabetic drugs.

Pharmacological Drug	Mechanism of Action	Genes	Main Effects	SNPs	Clinical Utility
Metformin	Inhibits gluconeogenesis through AMPK activation and other AMPK independent pathways	***SLC22A1*** (OCT1)	Located in the epithelial inner surface of the gut, renal tubules, and hepatocytes. OCT therefore allows the transport of the hydrophilic molecule across cell membrane barriers.	Highly polymorphic with up to 34 identified polymorphic SNPs across different populations. Among the common SNPs are: rs628031, rs622342, rs12208357, rs72552763, rs2297374, rs4646272, rs34130495, rs2282143, rs1867351, rs594709, rs200684404, *rs34104736*, *rs2297373*, *rs622591*, *rs2197296*, *rs4709400*, *rs461473*, *rs1443844*, *rs9457843*, *rs6937722*	Studies are highly controversial on the effect of polymorphism on the effectiveness and tolerance of metformin in different population cohorts
		***SLC22A2*** (OCT2)		*rs316019*, *rs316009*, and *rs145450955*	
		***SLC22A3*** (OCT3)		*rs3088442*, *rs2292334*, *rs12194182*, *rs543159*, *rs1317652*, and *rs2048327*	
		***SLC47A1*** (MATE1)	Renal efflux transporters promoting excretion	*rs2289669* and *rs8065082*	Possibly impact the degree of A1c reduction
		***ATM*** (AMPK?)	Promotes glucose uptake into skeletal muscles; represses lipogenesis and cholesterol biosynthesis	*rs1800058* and *rs11212617*	Possibly impact the degree of A1c reduction
Sulfonylurea	Increase in insulin secretion through ATP-dependent K channel inhibition	** *CYP2C9* **	Influences the clearance and plasma exposure of SU	CYP2C9*3 *rs1057910* and CYP2C9*2 *rs1799853*	Possibly associated with A1c and/or FBS reduction as well as increased risk of hypoglycemia
		** *KCNJ1* **	Subunits of the K_ATP_ channel that affects membrane polarization and insulin release	*rs5219*	Controversial
		** *ABCC8* **		*rs757110*	
		** *KCNQ1* **		*rs2237892* and *rs2237897*	
Thiazolidinediones	PPARγ agonists that enhance insulin sensitivity and glucose uptake	** *SLCO1B1* **	Transports the drug from blood to the liver and may contribute to in their metabolism	*521T*	Controversial
		** *PPARγ* **	peroxisome proliferator-activated receptor γ gene	*rs1801282*	
GLP-1 analogues	Stimulates glucose-dependent insulin release from pancreatic islets	** *CNR1* **	Cannabinoid type 1 receptor present in adipose tissue	*rs1049353*	No changes in glycemic outcomes, but changes in body weight are noted
		** *GLP1R* **	Glucose-dependent insulin release regulation. GLP1 RA genes also may contribute to delayed gastric emptying and increase satiety, while ARRB1 genes may affect beta cell proliferation	*rs6923761* *rs10305420*	Controversial, *rs6923761G* may be implicated in A1c reduction, or greater weight reduction and more marked delayed gastric emptying. while *rs10305420* is unlikely to be associated
		**TCF7L2**		*rs7903146*	Controversial
		**SORCS1**		*rs1416406*	Possibly affect glycemic control outcomes including A1c, fasting glucose and post-meal glucose
DPP4 inhibitors	DPP-4 inhibition promoting glucose regulation	** *GLP1R* **	GLP receptors	*rs6923761*, *rs3765467*	Possibly affect A1c reduction
		** *TCF7L2* **	Impairment in insulinotropic actions of incretin hormones, which are vital in DPP 4 inhibitor actions	*rs7903146*	Possibly affect A1c reduction
		** *PNPLA3* **	PNPLA3 gene has been associated with insulin resistance	*rs738409*	Possibly affect A1c reduction and PNPLA3 may additionally affect lipid profile and changes in liver aminotransferase
		** *DPP4* **	Drug inhibition site gene	*rs2909451*, *rs759717*	Likely higher DPP4 activity, may affect A1c reduction
		** *CDKAL1* **	Genotypes associated with higher DM incidence	*rs7754840*, *rs7756992*	Possibly affect A1c reduction
		** *KCN* **	Potassium gene family, that affects the release of insulin from pancreas	*rs734312*, *rs2285676*, *rs163184*	Possibly affect A1c reduction
		** *PRKD1* **	Kinase G-protein coupled receptor mediates insulin release	*rs57803087*	Possibly affect A1c reduction
SGLT2 inhibitors	Reduces renal tubular glucose reabsorption, therefore increasing glucose excretion	**SLC5A2**	Receptor gene for SGLT	*rs3116149*, *rs9934336*, *rs3813008*, *rs11646054*, *rs3116650*	Evidence to date shows no impact on clinical outcomes; likely negligible effect
		**UGT1A9**	Uridine diphosphate glucuronosyltransferase enzyme involved in metabolic elimination pathway	*rs72551330*	Higher exposure of the drug but unknown clinical impact; likely negligible effect

## Data Availability

Not applicable.

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
