# Peer review of "Harnessing Pharmacomultiomics for Precision Medicine in Diabetes: A Comprehensive Review"

_biomedicines, 2025, doi:10.3390/biomedicines13020447_

Round 1

Reviewer 1 Report

Comments and Suggestions for Authors

Authors comprehensively overviewed the potential of pharmaco-multiomics in advancing precision medicine for Type 2 Diabetes Mellitus. By emphasizing the integration of multiple omics data, the authors advocate for personalized treatment strategies tailored to the unique molecular profiles of individual patients.

The study effectively covers a wide range of omics disciplines; genomics, proteomics, metabolomics, transcriptomics, microbiomics, and epigenomics, highlighting their respective contributions to understanding T2DM and optimizing treatment.

The review underscores the importance of individualized therapy, a critical shift from the traditional "one-size-fits-all" approach in T2DM management. By exploring molecular-level variations, it presents a strong case for precision medicine.

Following issues must be addressed:

- Authors discussed the potential of multiomics approaches, but they did not provide concrete examples or case studies of their successful application in clinical settings. Including examples of precision medicine initiatives in T2DM would enhance the practical relevance of the paper.

- Integrating diverse omics data requires standardized methodologies and interoperable systems. A discussion on current efforts or future directions for standardizing data collection, storage, and analysis could strengthen the paper.

- The paper could benefit from a more detailed discussion of the challenges associated with implementing multiomics in clinical practice. Issues such as high costs, data integration complexity, and the need for specialized bioinformatics tools should be addressed.

Author Response

REVIEWER #1

Authors comprehensively overviewed the potential of pharmaco-multiomics in advancing precision medicine for Type 2 Diabetes Mellitus. By emphasizing the integration of multiple omics data, the authors advocate for personalized treatment strategies tailored to the unique molecular profiles of individual patients. The study effectively covers a wide range of omics disciplines; genomics, proteomics, metabolomics, transcriptomics, microbiomics, and epigenomics, highlighting their respective contributions to understanding T2DM and optimizing treatment. The review underscores the importance of individualized therapy, a critical shift from the traditional "one-size-fits-all" approach in T2DM management. By exploring molecular-level variations, it presents a strong case for precision medicine. Following issues must be addressed:

  1. Authors discussed the potential of multiomics approaches, but they did not provide concrete examples or case studies of their successful application in clinical settings. Including examples of precision medicine initiatives in 2DM would enhance the practical relevance of the paper.

Thank you for your insightful feedback. to address your comment, we have greatly expanded the discussion on pharmacogenomics and other omics approaches within the manuscript. For further details, please refer to lines 221-230 in page 6. Most notably, we added a substantial new section titled "Precision Medicine Initiatives in T2DM," where we discuss various international initiatives that exemplify the integration of pharmacogenomics and other multiomics data into clinical practices, including the Innovative Medicines Initiative (IMI) in Europe, the UK Biobank, the US Precision Medicine Initiative (PMI), and other national and international projects (Page 18, lines 628-660)

  1. Integrating diverse omics data requires standardized methodologies and interoperable systems. A discussion on current efforts or future directions for standardizing data collection, storage, and analysis could strengthen the paper.

We have added a new section to our manuscript titled "Standardizing Multi-Omics Data Integration in T2DM." Please refer to page 17, lines 594-625. This section discusses the current efforts and future directions for standardizing methodologies and improving the interoperability of systems for integrating diverse omics data.

  1. The paper could benefit from a more detailed discussion of the challenges associated with implementing multiomics in clinical practice. Issues such as high costs, data integration complexity, and the need for specialized bioinformatics tools should be addressed.

As you suggested, we have addressed these issues across several sections of our manuscript. In the standardizing muliomics data integration in T2DM paragraph spanning lines 614 to 625 in page 17, we discuss the logistical and financial challenges associated with implementing multiomics approaches. We explore how the high costs tied to these programs may affect their feasibility and scalability. In the conclusion section (Page 19, Lines 691-699), we provide a summary of the challenges related to the translation of this innovative approaches into clinical practice. Additionally, we have thoroughly reviewed our manuscript to highlight the specific omics challenges associated with T2DM.

Reviewer 2 Report

Comments and Suggestions for Authors

In this review article, the authors focused benefit and usefulness of integration of pharmco-multiomics data analysis for optimal individualized treatment in T2D. This is an important topic for the future advancement of diabetes treatment, but the following improvements are needed to disseminate knowledge to a wider audience.

1.     The introduction section is too wordy. It should be shorter and more to the point.

2.     In addition to Figure 1, two or three more schematic diagrams are needed to symbolically show the contents.

3.     The abbreviation of ADR in Figure 1 requires an explanation.

Author Response

REVIEWER #2

In this review article, the authors focused benefit and usefulness of integration of pharmco-multiomics data analysis for optimal individualized treatment in 2D. This is an important topic for the future advancement of diabetes treatment, but the following improvements are needed to disseminate knowledge to a wider audience.

  1. The introduction section is too wordy. It should be shorter and more to the point.

I appreciate you noting that this is an important topic for the future advancement of diabetes treatment. As you have suggested, we have thoroughly reviewed the text for repetitive statements and combined similar ideas to reduce redundancy and improve the flow of information. We also narrowed down the information to emphasize the most critical points. Additionally, we reduced the length of background details where possible, focusing on essential statistics and trends relevant to the upcoming sections of the paper. For further details, please refer to the revised sections spanning lines 40 to 179 in pages 1 to 4. Our efforts aim to streamline the content and enhance clarity without compromising the necessary context and background required for understanding the scope of our study on diabetes mellitus.

  1. In addition to Figure 1, two or three more schematic diagrams are needed to symbolically show the contents.

To address this point, we have made an additional figure 2 entitled “Integrative Omics Strategy in Precision Medicine.” This figure illustrates a framework for advancing precision medicine through the integration of clinical, molecular, social, and environmental data. It incorporates clinical and molecular data, including health history, demographics, genomics, and other omics sources, while employing an integrative omics approach to enhance data analysis for improved clinical outcomes and personalized treatments. Please refer to figure 2 and the legend (Page 20) for further details.

  1. The abbreviation of ADR in Figure 1 requires an explanation. 

As suggested, we have added a clear explanation for the abbreviation "ADR" within the legend of Figure 1. Additionally, we included context about adverse drug reactions (ADRs), please refer to  lines 94-97, page 2 and lines 160-162, page 4.

Reviewer 3 Report

Comments and Suggestions for Authors

Dear Authors,

the comments in the annex file.

Best

Comments on the Quality of English Language

The English merit attention

Author Response

REVIEWER #3

First of all, thank you for providing me with the opportunity to enhance my skills through your valuable manuscript. After a careful reading, however, I must point out several aspects, particularly in the methodological section, which, in my humble opinion, deserve more attention before submission to such a prestigious journal:

  1. The title does not clearly convey (and honestly, neither does other parts of the manuscript) the type of study conducted, which would certainly highlight the research more effectively and assist a careful reader in finding the appropriate references when conducting a search or review.

We sincerely appreciate your valuable feedback. According to your observations, we have revised the title to more accurately reflect the nature and focus of the study. The new title is: "Harnessing Pharmaco-Multiomics for Precision Medicine in Diabetes: A Comprehensive Review." I hope with this new title we emphasize the use of multi-omics approaches in advancing precision medicine for diabetes. Furthermore, we have refined the introduction and conclusion to ensure a consistent emphasis on the review nature of the study, thereby enhancing clarity and making it easier for readers to identify the scope and contributions of our work during searches or reviews.

  1. The Abstract (as well as the Introduction and Discussion sections) lacks a clear practical implication of the results obtained. This is essential in order to provide more scientific support to the manuscript. The abstract is somewhat verbose and generic; it deserves more attention.

We have thoroughly revised the whole manuscript as you have suggested. We emphasize the transformative potential of integrating multi-omics data to personalize treatment strategies for T2DM. The introduction now includes a more focused discussion on the need for precision medicine frameworks in addressing the heterogeneity of T2DM. All changes and additions are made with track changes.

  1. The Introduction is unclear and lacks a focused presentation of the study topic, with no clearly stated objective for the study (or at least one that is difficult to infer by a careful reader).

We have made several key revisions to ensure clarity and focus in the introduction. We also reviewed the text for repetitive statements, combining similar ideas to reduce redundancy and improve the flow of information. The introduction now presents a clearer objective, highlighting the necessity of precision medicine frameworks to address the heterogeneity of T2DM. These changes aim to provide a cohesive narrative that enhances reader comprehension of the study's scope and objectives. For further details, please refer to the revised introduction in pages 1 to 4.

  1. There is a lack of a clearly defined methodology to summarize the results presented—was this a narrative review?

We appreciate your observation regarding the methodology.  Our study is a narrative review, which involves synthesizing and discussing existing literature on multi-omics approaches in precision medicine for T2DM. As you suggested, we have now stated this in the methodology section, outlining our approach in selecting and analyzing relevant studies to provide a comprehensive overview of the topic. Major revisions have been made throughout the manuscript to ensure clarity and alignment with our narrative review approach.

  1. The limitations of the study are missing, which should definitely be addressed in a dedicated section, as they are very important for the type of study conducted. The discussion section is a mix with the results; I recommend a more concise and focused discussion specifically on the topic of the study.

To address this point, we have revised the whole manuscript to include the limitations of the study.  For further details, please refer to the revised section spanning lines 661 to 682 on pages 18 and 19.

  1. Some references need to be improved or updated, as many are quite outdated, which limits the proper interpretation of the results obtained, especially in relation to the study objectives that, moreover, are never clearly stated in the text.

Thank you for this insightful remark. As you suggested, outdated references have been replaced with recent, relevant studies, particularly those published within the last five years. These updates are integrated throughout the whole manuscript to provide a more robust interpretation of the findings, situating them within the context of current advancements and highlighting their significance to the field. Additionally, we took the opportunity to thoroughly check all references once again, including the newly added and updated citations, to ensure their accuracy and relevance.

Round 2

Reviewer 2 Report

Comments and Suggestions for Authors

I have no further comment.

Author Response

Thank you for your feedback. We appreciate your guidance throughout this process.

Reviewer 3 Report

Comments and Suggestions for Authors

Dear Authors,

the comments in the annex file.

Best.

gc

Author Response

Thank you for your feedback and for acknowledging the significant improvements made to the manuscript. We update the terminology for type 2 diabetes mellitus to "type 2 diabetes" (T2D) as per your suggestion. We appreciate your guidance throughout this process.